# Advanced Optical Microscopy: Unveiling Functional Insights Regarding a Novel *PPP2R1A* Variant and Its Unreported Phenotype

**DOI:** 10.3390/ijms241813699

**Published:** 2023-09-05

**Authors:** Mònica Roldán, Gregorio Alexander Nolasco, Lluís Armengol, Marcos Frías, Marta Morell, Manel García-Aragonés, Florencia Epifani, Jordi Muchart, María Luisa Ramírez-Almaraz, Loreto Martorell, Cristina Hernando-Davalillo, Roser Urreizti, Mercedes Serrano

**Affiliations:** 1Confocal Microscopy and Cellular Imaging Unit, Genetic and Molecular Medicine Department, Pediatric Institute for Rare Diseases, Hospital Sant Joan de Déu, 08950 Barcelona, Spain; monica.roldan@sjd.es (M.R.); marcos.frias@sjd.es (M.F.); 2Institut de Recerca Sant Joan de Déu, 08950 Barcelona, Spain; alexander.nolasco@sjd.es (G.A.N.); florencia.epifani@sjd.es (F.E.); jordi.muchart@sjd.es (J.M.); loreto.martorell@sjd.es (L.M.); cristina.hernando@sjd.es (C.H.-D.); roser.urreizti@sjd.es (R.U.); 3Pediatric Neurology Department, Hospital Sant Joan de Déu, 08950 Barcelona, Spain; 4Quantitative Genomic Medicine Laboratories, qGenomics, 08950 Barcelona, Spain; lluis.armengol@qgenomics.com (L.A.); marta.morell@qgenomics.com (M.M.); manel.garcia@qgenomics.com (M.G.-A.); 5Diagnostic Imaging Department, Hospital Sant Joan de Déu, 08950 Barcelona, Spain; 6Genetic and Molecular Medicine Department, Pediatric Institute for Rare Diseases, Hospital Sant Joan de Déu, 08950 Barcelona, Spain; marialuisa.ramirez@sjd.es; 7Centro de Investigación Biomédica en Red de Enfermedades Raras (CIBER-ER), Instituto de Salud Carlos III, 28220 Barcelona, Spain; 8Clinical Biochemistry Department, Hospital Sant Joan de Deu, 08950 Barcelona, Spain

**Keywords:** functional studies, cerebellar atrophy, confocal microscopy, neurodevelopmental disorders, pontocerebellar hypoplasia, *PPP2R1A*, super-resolution microscopy

## Abstract

The number of genes implicated in neurodevelopmental conditions is rapidly growing. Recently, variants in *PPP2R1A* have been associated with syndromic intellectual disability and a consistent, but still expanding, phenotype. The *PPP2R1A* gene encodes a protein subunit of the serine/threonine protein phosphatase 2A enzyme, which plays a critical role in cellular function. We report an individual showing pontocerebellar hypoplasia (PCH), microcephaly, optic and peripheral nerve abnormalities, and an absence of typical features like epilepsy and an abnormal corpus callosum. He bears an unreported variant in an atypical region of *PPP2R1A*. In silico studies, functional analysis using immunofluorescence, and super-resolution microscopy techniques were performed to investigate the pathogenicity of the variant. This analysis involved a comparative analysis of the patient’s fibroblasts with both healthy control cells and cells from an individual with the previously described phenotype. The results showed reduced expression of *PPP2R1A* and the presence of aberrant protein aggregates in the patient’s fibroblasts, supporting the pathogenicity of the variant. These findings suggest a potential association between *PPP2R1A* variants and PCH, expanding the clinical spectrum of *PPP2R1A*-related neurodevelopmental disorder. Further studies and descriptions of additional patients are needed to fully understand the genotype–phenotype correlation and the underlying mechanisms of this novel phenotype.

## 1. Introduction

A growing body of research indicates a significant increase in the number of genes implicated in neurodevelopmental conditions altering several molecular pathways that involve various cellular processes and signaling networks that play critical roles in brain development and function. Although some serine/threonine protein phosphatases play a critical role in the control of cell function, they have barely been associated with developmental disorders [1]. Within the cell, there are two major families of serine/threonine (Ser/Thr) protein phosphatases: protein phosphatase 1 (PP1) and protein phosphatase 2A (PP2A). PP2A is a heterotrimeric enzyme [2,3], composed of a catalytic subunit (C), a substrate-binding regulatory subunit (B), and a scaffolding subunit (A) that links the regulatory and catalytic subunits. The scaffolding A and the regulatory B subunits are encoded by the *PPP2R1A* and *PPP2R5D* genes, respectively [2,3]. 

Recently, associations between de novo missense pathogenic variants of PPP2R5D, PPP2R1A, and PPP2CA have been reported, leading to autosomal-dominant forms of intellectual disability (MRD35 (OMIM#616355), MRD36 (OMIM#616362), and NDLBA (OMIM#618354), respectively, also known as Houge-Janssen syndrome 1,2, and 3) [1,4]. In 2015, in the context of the Deciphering Developmental Disorders (DDD) study, five de novo *PPP2R1A* mutations were identified among 1133 parent–child trios [1]. The PPP2R1A protein is composed of 15 HEAT (huntingtin, elongation factor 3, protein phosphatase 2A, and yeast kinase TOR1) repeat motifs, of which HEATs 1–8 mediate interactions with a specific regulatory B subunit [4]. Among the 46 patients reported to date, forty-five showed pathogenic variants that accumulate in HEATs 4–7 [1,5,6,7,8,9,10], generating a dominant negative effect, and causing biochemical dysfunction in the majority of them [1,7]. The subjects reported with *PPP2R1A*-related neurodevelopmental disorder (NDD) show a consistent neurological phenotype (corpus callosum hypoplasia, epilepsy, moderate-to-severe intellectual disability, and ventriculomegaly) but the clinical spectrum is expanding and extraneurological features, such as congenital heart disease, have recently been reported [8]. Remarkably, in two patients bearing the same *PPP2R1A* variants at HEAT5, in the presence of severe ventriculomegaly and a very extensively affected central nervous system, abnormal posterior fossa structures were observed, showing cerebellar and brainstem hypoplasia [5,8]. Both infants died prematurely and showed associated perinatal epilepsy. However, isolated pontocerebellar hypoplasia (PCH) as a cardinal sign has not been reported. 

PCH comprises a group of clinically and genetically heterogeneous rare neurodegenerative disorders with a prenatal onset, characterized by abnormal growth and survival of neurons in the cerebellum, inferior olives, and ventral pons. Radiologically and pathologically, all PCH subtypes are characterized by hypoplasia and variable atrophy of the cerebellum and pons. Currently, pathogenic variants in at least 19 different genes explain different clinical subtypes of PCH based on distinct clinical, biochemical, and radiological features [11,12,13,14,15]. Proteins encoded by PCH-associated genes are involved in various cellular functions, predominantly RNA metabolism and protein expression, but also nucleotide metabolism, mitochondrial function, and vesicular transport [13,14,15]. Individuals affected by PCH share clinical features, such as prenatal or perinatal onset of disease, progressive microcephaly and global developmental delay with severe intellectual and motor function impairments, epilepsy, and, frequently, death during childhood. However, recent subtypes have been related to a mild, non-degenerative disease course [16]. To date, no mutations in the protein phosphatase have been reported as causing PCH. 

The present study aims to report a novel variant in a region of *PPP2R1A* where mutations have not been reported so far, leading to an undescribed phenotype, which shows PCH, congenital microcephaly, optic and peripheral nerve abnormalities, and an absence of some expected features such as epilepsy and an abnormal corpus callosum. To further establish the pathogenicity of the variant, comprehensive investigations were conducted, including in silico studies, confocal microscopy, and computational super-resolution microscopy functional studies. These involved a comparative analysis of the patient’s fibroblasts with both healthy control cells and cells from an individual with the previously described phenotype. 

## 2. Results

### 2.1. Clinical Report and Molecular Findings

Subject 1 (S1) was born of healthy unrelated Caucasian parents. Pregnancy was uneventful and he was born at term showing microcephaly (PC 32 cm (2nd percentile, −2.1 standard deviation (SD)) and normal weight and length (weight of 2760 g (6th percentile, −1.63 SD), length 48 cm (8th percentile, −1.46 SD). The perinatal period was normal. During the first months of life, he showed global developmental delay and hypotonia, with normal growth but maintaining the microcephaly.

From early infancy, he showed global psychomotor deficits, affecting gross and fine motor skills, as well as language development retardation. He showed convergent strabismus. At the age of 4 years, he achieved independent walking, though he exhibited an ataxic, wide-based gait. He was referred to our hospital at the age of 6 years and first magnetic resonance imaging (MRI) showed pontine and cerebellar asymmetric hypoplasia, and an incipient increase in the cerebellar interfoliar spaces (Figure 1A). There was a ventricular enlargement without signs of hydrocephalus. By that age, he had attended special schooling due to moderate difficulties in understanding, speech difficulties using simple sentences, and motor clumsiness with an ataxic gait. In the physical exam, signs of spasticity, including increased deep tendon reflexes (DTR) and shortened Achilles’ tendons, were observed, as well as the previously mentioned convergent strabismus and horizontal nystagmus. A first nerve conduction study at the age of 10 years revealed mild neuroaxonopathy that remained stable during evolution. To improve his gait, he underwent orthopedic surgery on both Achilles’ tendons. Cranial MRIs at 10 and 16 years, respectively, showed pontocerebellar atrophy progression and optic nerve atrophy (Figure 1B,C), which were not previously detected in the neuroimaging and were not related to visual impairment. The supratentorial structures were normal, except for microcephaly, without any structural anatomical abnormalities, and the corpus callosum was normal. At age 13, a cognitive evaluation yielded an intelligence quotient score of 41 (Wechsler intelligence scale for children, WISC-IV). He never presented epilepsy or febrile seizures and, during evolution, there were no other organ or system comorbidities.

At the age of 18, he is able to follow simple orders, and his expressive language is simple and characterized by slurred speech, but he is able to maintain a conversation easily. He has intermittent horizontal nystagmus and subtle strabismus with papillary atrophy on funduscopy. His weight and height are normal (66.4 kg (−0.72 SD) and 178 cm (0.09 SD), respectively). No dysmorphic traits are evident, other than the microcephaly (53 cm, 2nd percentile, −2.20 SD). He can walk unassisted, and he presents subtle intention tremor and dysdiadokokinesis, predominantly on the non-dominant side, as well as mild truncal ataxia. Cerebellar assessment using the International Rating Cerebellar Ataxia Score resulted in a score of 21 out of 100 [17]. Following the proposal by Lanaerts et al. [7] of a severity score for *PPP2R1A*-related NDD individuals, S1 obtained a three-point score. 

Following suspicion of PCH, a gene panel analysis was performed, yielding normal results. Additionally, chromosomal microarray analysis was performed, yielding normal results, as well. Further investigation through WES (whole-exome sequencing) trio analysis identified an unreported heterozygous de novo missense variant in the *PPP2R1A* gene (19:52693410 C > T; ENST00000322088.6/NM_014225.6; c.61C > T, p.Arg21Cys). This variant is absent in the gnomAD database and is predicted to be deleterious based on in silico predictors (CADD 25; DANN 0.998; SIFT; and Polyphen2, pathogenic). This variant has been previously reported in ClinVar (ID: 2288085) as a variant of unknown significance (VUS) but the clinical status of the proband and inheritance is not specified. The *PPP2R1A* gene is strongly constrained for missense variants according to gnomAD v2 (Z = 4.46). Following the ACMG classification guidelines [18], this variant is classified as pathogenic (matching criteria: PS2 PS3 PM2 PP2). Despite an extensive search of all genes related to PCH to date, no other plausible mutations were found. Furthermore, in order to explore potential mutations associated with disorders causing cerebellar atrophy, particularly early-presentation spinocerebellar ataxia (SCA), a panel of expansion mutations was also performed, ruling out expansions.

Subject 2 (S2) is a 12-year-old Caucasian girl, the daughter of healthy non-consanguineous parents, who was included in a recent report [7]. During pregnancy, a prenatal ultrasound identified intrauterine growth retardation (IUGR) in the third trimester. Delivery was uneventful at 37 weeks of gestation, with low weight (1820 gr, −2.38 SD) but normal length and head circumference. Global developmental delay was identified from six months of age when a physical exam revealed axial hypotonia, joint laxity, a high palate, strabismus, and a long face. On physical examination, at 12 years old, she presented microcephaly of 50 cm (1st percentile, −2.9SD), with a long face, large maxillary central incisors, a narrow upper palate, left-divergent strabismus, and no nystagmus. Language is scarce but functional. She presents joint laxity, scoliosis, hypotonia, and increased DTR in the lower limbs. She attends special schooling and presents intellectual disability, learning difficulties, and a lack of cognitive flexibility with rigid behavior. A brain MRI at 3 years revealed hippocampal dysplasia and no other abnormal findings. Nerve conduction and electromyographic studies showed no abnormalities. She never presented epilepsy or other organ involvement. S2′s WES trio identified a heterozygous de novo pathogenic missense variant in *PPP2R1A* (c.539T > G, p.Met180Arg), previously reported in the literature. Applying the severity score for *PPP2R1A*-related NDD individuals, S2 obtained a four-point score [7].

### 2.2. In Silico Studies

The *PPP2R1A* p.Arg21Cys change affects the same PP2A subunit B binding region (spanning residues 8-399) as the previously described pathogenic mutations (Figure 2 and Appendix A). This variant is located in the HEAT1 repeat, while most previously published variants lay within HEAT repeats 4 to 7, and most frequently in HEAT 5 (Figure 2 and Appendix A). This variant is, hitherto, the most N-terminal pathogenic variant described so far. A conservation study of residue Arg21 shows that it is fully conserved among vertebrates (Appendix A). In silico studies of the tertiary structure of HEATs 1 and 2 show that Arg21 putatively forms hydrogen bonds with the nearby Thr58 (Appendix A).

### 2.3. Functional Analysis Using Confocal and Super-Resolution Imaging Techniques

Immunofluorescent labeling of PPP2R1A in the subjects’ fibroblasts showed varying staining intensities (Figure 3). A lower signal intensity corresponding to *PPP2R1A* expression was observed in S1 (our case of interest, bearing the new variant p.Arg21Cys) than in S2 (carrying the previously reported p.Met180Arg variant), and the control.

A decreased intensity signal was found in both the nucleus and the whole cells of patient S1 whereas S2′s cells presented closer values to the control (Figure 3). The fluorescence intensity of PPP2R1A did not conform to a normal distribution. There were noteworthy variations observed among the three groups (S1, S2, and control). Furthermore, in pairwise comparisons, significant differences were detected between the two patients and the control. Similarly, when analyzing the fluorescence of PPP2R1A in the fibroblast nucleus, more substantial differences were identified between the groups.

Computational super-resolution microscopy revealed the presence of aberrant protein aggregates in the cytoplasm, supporting the pathogenicity of the finding (Figure 4). A representative 3D volume of the PPP2R1A protein in S1, S2, and the control was generated and is visualized in Figure 4A,B. Additionally, Figure 4C illustrates the quantification of both the number and area of protein aggregates. A remarkable variation in aggregate size was observed in S1, ranging from less than 0.001 µm2 to aggregates larger than 55 µm2 (Figure 4). In the index case, the aggregates exhibited a similar range of sizes, but with less variability (0.1–42.1 µm2). In comparison, the control group displayed signals with smaller aggregates (0.1–14 µm2) (Figure 4). Significant differences existed between the two groups under study. Furthermore, in the pairwise comparisons, significant differences were also observed between the two affected subjects and the control sample.

## 3. Discussion

The neurodevelopmental disorders related to PP2A dysfunction encompass a group of overlapping syndromes characterized by severe PP2A dysfunction. The phenotypic features related to the described neurodevelopmental syndrome due to variants in *PPP2R1A* are neurodevelopmental and language delay, hypotonia, behavior problems, frontal bossing, a long face, joint hypermobility, and hypoplasia or agenesis of the corpus callosum, which are present in more than 50% of patients [7]. 

To date, two reports describe subjects with variants in *PPP2R1A* and associated PCH. The first one was reported by Wallace et al. [5] describing an infant bearing a previously reported *PPP2R1A* variant (c.548G > A; p.Arg183Gln) with severe ventriculomegaly and severe neuroanatomical distortion of both supratentorial and infratentorial structures and presenting epilepsy during the first days of life. Baker et al. reported the second one [8], also presenting with severe ventriculomegaly, corpus callosum agenesis, and hypoplastic brainstem with vermian hypoplasia. This second baby showed the same variant and presented with epilepsy during the first days of life, too. In fact, p.Arg183Gln has been associated with the most severe phenotype. Additionally, in a murine model, it has been associated with the most profound binding deficiency [20], as also found in the biochemical characterization reported by Lenaerts et al. [7]. Both subjects presented very premature death, and posterior fossa structure abnormalities at least partly affected by the severe ventriculomegaly. Unfortunately, a third patient with ventriculomegaly and posterior fossa abnormalities was incompletely described, since the pathogenic variant in *PPP2R1A* is not detailed at all [10]. In the present study, the S1 proband presented with isolated PCH on neuroimaging, pointing towards an initial malformative rhombencephalic basis and not a secondary disruptive abnormality, in the context of a generalized central nervous system alteration. Moreover, optic nerve atrophy and peripheral nerve implications have never been reported in relation to *PPP2R1A*. 

Cerebellar hypoplasia and atrophy are common in neuropediatrics, occurring in a very heterogeneous group of disorders and frequently as a nonspecific finding. However, the coexistence of midbrain hypoplasia narrows down the diagnostic options. Initially, since S1 presented with microcephaly and PCH, and in the absence of a sentinel event in their personal history explaining a possible non-genetic but disruptive cause, the diagnostic search was focused on classical PCH-related genes, without success. From a clinical perspective, the most similar PCH type is PCH3, which has been particularly associated with optic nerve atrophy and an increase in DTR. However, the molecular basis of PCH3 is biallelic variants in the *PCLO* gene, encoding a protein related to the regulation of vesicle formation and synaptic vesicle trafficking, but the sequencing of *PCLO* was normal in S1. PCH or neurogenetic disorders causing pontocerebellar hypoplasia are a growing group of genetic conditions [11], and new genes or new phenotypes related to already known genes will probably be discovered in the near future. Since the finding of a *PPP2R1A* variant in S1 was unexpected, a more detailed revision of the sequences of all those genes related to PCH and diseases associating cerebellar and brainstem hypoplasia was performed. Moreover, studies of repeat expansion mutations were also performed, including early-onset SCAs. All other filtered variants found in the WES were carefully reviewed, in an attempt to rule out any contribution explaining or distorting the phenotype.

All previously reported pathogenic variants in *PPP2R1A* (seventeen to date) are clustered close to the regulatory subunit-binding region of the protein, accumulating in HEATs 4–7 [1,4,5,6,7,8,9] (Figure 2, Appendix A) and causing single-amino-acid substitution. These variants generate a dominant negative effect and there is compelling evidence that the severity of the phenotypic spectrum correlates with biochemical dysfunctions [7]. S1 presented a missense mutation outside of the reported regions, leading to a replacement of the amino acid arginine (basic side chain, positively charged) with the amino acid cysteine (polar, uncharged side chain) at residue 21. The high evolutionary conservation of this altered residue supports its pathogenicity. Notably, the whole *PPP2R1A* protein sequence shows high conservation, especially among vertebrata and, for this particular residue, in other organisms, lysine (K), also basic and positively charged, is found. Moreover, in silico analysis of previous variants and the Arg21Cys mutation showed that it is located closer to the catalytic subunit, which is certainly outside of the described cluster of mutations, which are closer to the regulatory subunit. These in silico studies also show that Arg21 seems to form hydrogen bonds with the HEAT2 alpha helix, and could play a role in 3D structure maintenance. These bonds are lost in the presence of cysteine, suggesting that this change can affect protein folding, which could be in agreement with the increase in protein aggregates observed in the S1 patient’s fibroblasts. Moreover, in silico analysis of previous variants and the S1 mutation showed that it is located close to the catalytic subunit, which is certainly outside of the described cluster of mutations, which are closer to the regulatory subunit. This atypical localization may explain the unexpected clinical phenotype, but in silico studies are not enough to support pathogenicity. In this context, functional studies currently play a crucial role in establishing the connection between new variants and emerging phenotypes. Immunofluorescent labeling denoting the expression of *PPP2R1A* in the cytoplasm depicts the abundance of PPP2R1A, which was clearly greater in the control cells than in S2, and in turn, greater than S1, bearing the unreported variant. 

Computational super-resolution microscopy revealed the presence of aberrant protein aggregates in the cytoplasm of S1′s fibroblasts, abnormal due to their size (moderate and large aggregates) but also due to the huge variability in size. When they were quantified, progression among the cells of S1, S2, and the control was identified, showing that S1′s cells present greater abnormalities than S2′s cells, which showed, in turn, greater abnormalities than the control cells. Taken together, these findings point to a greater distortion or cellular effect related to the new variant. In summary, from a cellular point of view, greater abnormalities have been demonstrated in S1 than in S2, who presents the already known phenotype. Although global severity measured using Lanaerts’ score is similar for both [7], S1 also has peripheral nerve implications and optic nerve atrophy, which were not included in the proposed score design.

Regarding the biological significance of protein aggregates, they are subcellular perinuclear structures in which misfolded proteins accumulate through retrograde transport on microtubules [21], and highly dynamic processes are involved in their formation [22]. The presence of relatively large protein aggregates in any cell type indicates an impairment of the protein elimination machinery, compromising proteostatic homeostasis and leading to cellular stress and dysfunction. Additionally, the fact that the aggregates are relatively large affects the localization and movement/dynamics of other proteins and cellular organelles (differences compared to the control might be found if we were to study different cellular organelles). This is likely to result in the disruption of cellular processes, particularly intracellular transport (which can have significant implications at the neurological level). Furthermore, the presence of aggregates that can affect cellular organelles will also induce oxidative stress on proteins, lipids, and nucleic acids, triggering specific cellular responses that may ultimately lead to inflammatory reactions during the course of the disease. In general, the presence of large aggresomes in cells can trigger a series of cellular responses and compensatory mechanisms to counteract their detrimental effects. However, if these responses are insufficient, or if aggresomes continue to accumulate, they can contribute to the deterioration of cellular function and may be associated with neurodegenerative diseases [23] such as PCH, originally reported as a disorder of prenatal onset neurodegeneration, and as observed, from a neuroradiological point of view, in S1, showing greater cerebellar atrophy with age. Interestingly, S1 has not presented any regression or worsening from a clinical point of view. This is like some newly reported PCH types that behave in a stable manner, despite the common rule that the degenerative nature of PHC frequently leads to premature death [11,12,13,14,15]. 

It is important to consider that increased cellular resistance to proteotoxicity in patients may not be due solely to a lower level of protein aggregates, but rather, to the more efficient management of protein aggregates by the cell through the formation of inclusion bodies resembling aggresomes [23], as may be the case with our patient S1.

Although the functioning of PP2A is not well understood, it is well established as a regulator of cell division, growth, and differentiation, and the functions of PP2A in cancer and various neurodegenerative disorders, such as Alzheimer’s disease, have been studied in detail; for PPP2R1A, the crucial mechanism is determined by an alteration in the dependent dephosphorylation dynamics [24].

Neurological disease diagnosis has progressively evolved towards a molecular definition, but gene characterization still presents a challenge, which complicates the diagnostic process. Advances in Next-Generation Sequencing (NGS) techniques have greatly facilitated the identification of novel phenotypes associated with previously known genes and new variants in genes, shedding light on previously unreported phenotypes. Currently, functional studies play a crucial role in establishing the connection between new variants and emerging phenotypes. Although advanced optical microscopy techniques have been used extensively in research, their potential in diagnostic applications is yet to be fully explored. Their speed, ease of use, simple sample preparation, and relatively affordable instrumentation compared to electron microscopy make them potentially more accessible to clinicians and suitable for routine diagnosis. Collaboration between clinicians and researchers performing exhaustive phenotyping, NGS, and functional validations may prove essential to uncovering new molecular causes of PCH, as well as new phenotypes related to recently reported pathogenic genes, such as *PPP2R1A.*

To summarize, hypoplasia of the cerebellum and the brainstem has been only rarely associated with *PPP2R1A*-associated NDD. When present, it has been linked to the most severe phenotype, presenting a recurrent variant. We report an individual with a novel *PPP2R1A* variant and PCH in the context of a stable clinical condition, and not showing other typical features of *PPP2R1A*-associated NDD, while presenting optic atrophy and peripheral neuropathy. These findings suggest a potential association between *PPP2R1A* variants and PCH, expanding the clinical spectrum of neurodevelopmental disorders associated with *PPP2R1A*. Further studies and descriptions of additional patients are needed to improve the understanding of the genotype–phenotype correlation and the underlying mechanisms of this novel phenotype.

## 4. Materials and Methods

### 4.1. Phenotype and Molecular Studies

Two individuals bearing variants in *PPP2R1A* were studied. The first subject (S1) presented a variant in a not-typical region of *PPP2R1A*, while the second subject (S2) had a variant in the region where previous variants have been reported and her clinical features were recently published [7]. Exhaustive phenotyping was performed, including studies of dysmorphology and general and neurological exams. For Subject 1 (S1), conventional MRI (1.5 T scanner, GE Healthcare, Milwaukee, WI, USA) was performed at the ages of 6 years, 10 years, and 16 years. A nerve conduction velocity (NCV) study was performed at the age of 10 years via standard techniques using a Nantus Key Point Focus electromyograph with surface electrode recordings. 

Blood samples were collected from both individuals and their parents, after obtaining written informed consent. Both individuals underwent chromosomal microarray analysis. Additionally, for S1, PCH panel gene, trio-based WES and testing for nucleotide repeat expansions (to rule out other potential variants in early-presenting spinocerebellar ataxias genes (SCA)) were performed. For S2, after normal chromosomal microarray analysis, a WES trio was performed.

### 4.2. In Silico Studies

In silico modeling of the multimeric protein was performed using the PyMol Molecular Graphics System (v.2.4.1, Schrödinger, LLC, 1540 Broadway, NY, USA) with the Protein Phosphatase 2A Holoenzyme model (2NPP) [18]. For conservation analysis, human *PPP2R1A* was used as a BLAST query and model organisms were selected. Multiple protein alignment was performed using Clustal Omega, EMBL-EBI ver. 1.2.4 https://www.ebi.ac.uk/Tools/msa/clustalo/ accessed on 9 April 2023.

### 4.3. Functional Studies

Immunofluorescent labeling of *PPP2R1A* in the patients’ fibroblasts to locate and quantify this protein was performed. Fibroblasts from a healthy control (12-year-old boy) were obtained from the Sant Joan de Déu Barcelona Children’s Hospital Biobank. These cells were cultured in Dulbecco’s modified Eagle medium (Sigma-Aldrich, St. Louis, MO, USA) supplemented with 10% *v*/*v* fetal bovine serum (FBS, Gibco, Thermo Fisher Scientific, Inc., Waltham, MA, USA), 2 mmol/L L-glutamine (Sigma-Aldrich, St. Louis, MO, USA), and 100 mg/mL penicillin–streptomycin (Sigma-Aldrich, St. Louis, MO, USA). Cell cultures were maintained at 37 °C in a 5% CO_2_ humidified atmosphere. For immunofluorescence studies, fibroblasts were seeded onto glass coverslips and fixed in 4% formaldehyde for 20 min. Permeabilization was performed using 0.5% TritonX-100 diluted in PBS for 30 min at RT. Cells were blocked in PBS and 6% BSA for 45 min at RT. The anti-PPP2R1A primary antibody (1:100) was used and incubated for 1 h at 37 °C (ab154551, Abcam, Waltham, MA, USA). 

After washing in PBS, cells were incubated at 37 °C for 1 h in the dark using the secondary antibody α-rabbit Alexa Fluor 594 (A21207, Thermo Fisher Scientific, Inc., Waltham, MA, USA). Nuclei were stained with Hoechst 33342 trihydrochloride trihydrate (Life Technologies, Carlsbad, CA, USA), and then, mounted with Prolong Diamond antifade (Life Technologies). Confocal and super-resolution microscopy analysis was performed using a Leica TCS SP8 equipped with a white light laser, a HyVolution super-resolution module, and hybrid spectral detectors (Leica Microsystems GmbH, Mannheim, Germany). For the quantification, confocal images of the fibroblast cultures were acquired using an HC × PL APO 63×/1.4 oil immersion objective. Hoechst 33342 was excited using a blue diode laser (405 nm) and detected in the 420–475 nm range. *PPP2R1A* was excited using a white light laser (594 nm) and detected in the 635–795 nm range. An image sequence (XYZ) comprising 10 sections with a step size of 0.7 μm was captured to visualize the fluorescence distribution of the *PPP2R1A* protein throughout the cell’s thickness. A total of 122 fields were acquired, each with a field of view (FOV) measuring 184.52 × 184.52 μm. To represent the variation in intensity for each spectral component, a 12-bit encoding scheme was employed.

For precise data collection, we implemented appropriate negative controls to calibrate the confocal settings and eliminate non-specific fluorescence artifacts. To ensure comparability across different samples, identical confocal settings were maintained during image acquisition. Subsequent to image capture, we carried out sum intensity projections and fluorescence quantification using the ImageJ Fiji software (Image J 1.52n) (National Institutes of Health in Bethesda, MD, USA). The mean fluorescence intensity was then normalized by dividing it by the total number of cells. In the case of nuclei, a mask was generated using the nucleus staining channel, and the mean fluorescence intensity for each nucleus was quantified. 

Super-resolution images were acquired using an HC × PL APO 100x/1.4 oil immersion objective, an HyD detector, and mode HyVolution, with the pinhole set to 0.6 Airy units. To investigate the distribution of PPP2R1A in three dimensions, Z stacks were acquired at 0.2 μm intervals throughout the cell thickness, resulting in a total of 28 sections. These images had a resolution of 0.071 × 0.071 µm. Image deconvolution was performed using Huygens Professional software v17.10.0p7 64b (SVI, Leiden, The Netherlands) and stacks were reconstructed and visualized as three-dimensional (3D) volumes using Imaris software ver. 7.2.1 (Bitplane AG, Zürich, Switzerland) (Bitplane, Zürich, Switzerland). *PPP2R1A* aggregate area analysis was conducted using Imaris software (Surface module), and the aggregate areas were measured using the Surface Statistics function.

### 4.4. Statistical Analysis

The data are presented as means ± SEM and are displayed either as column bars or scatter plots, with error bars. *p*-values are indicated by asterisks as follows: * *p* < 0.05, ** *p* < 0.01, *** *p* < 0.001, and **** *p* < 0.0001. Graphs were created and statistical analysis was performed using GraphPad Prism version 8.0.1 (GraphPad Software, Inc., La Jolla, CA, USA).

The data’s normality were assessed using a Kolmogorov–Smirnov test, which indicated that the data did not conform to a normal distribution. Consequently, a Kruskal–Wallis test was utilized to compare all samples, while a Mann–Whitney test was performed to compare the control data with individual patient data, both in pairwise comparisons and across patients. The significance levels are indicated as * *p* < 0.05, ** *p* < 0.01, *** *p* < 0.001, and **** *p* < 0.0001.

### 4.5. Ethical Issues

Ethical permission for the study was obtained from the Research and Ethics Committee of the SJD Research Foundation. Parents gave their written informed consent and children/adolescents and adults gave their assent. Samples were obtained in accordance with the Helsinki Declaration of 1964, as revised in October 2013 in Fortaleza, Brazil. 

## Figures and Tables

**Figure 1 ijms-24-13699-f001:**
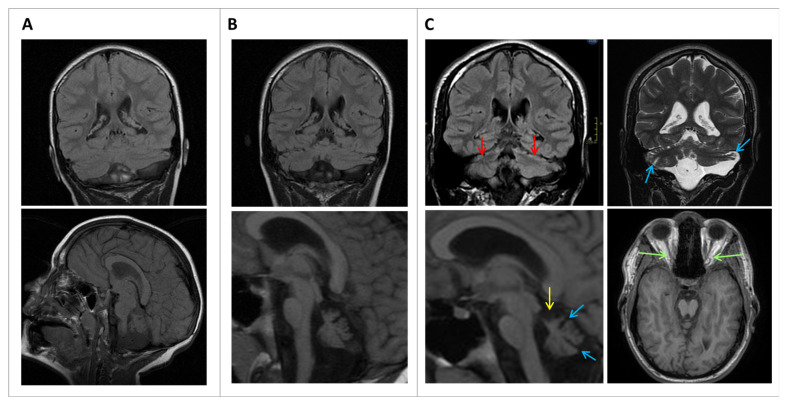
MRI at 6 years ((**A**): Coronal FLAIR T2, Sagittal FSE T1), at 10 years (**B**): Coronal FLAIR Details of image amplification have been notedT2, Sagittal FSE T1 amplified ×2), and 16 years of age (**C**): Coronal FLAIR T2, Coronal FSE T2, Sagittal FSE T1 amplified ×2.5, Axial 3D T1 SPGR). The initial MRI (**A**) shows the asymmetric volume of the cerebellar hemispheres, with the left hemisphere being smaller. There is also prominent inferior CSF space and atrophy of the vermis. No corpus callosum abnormalities are found. Follow-up MRIs (**B**,**C**) show progressive atrophy of both the cerebellum and the brainstem. There is a subtle widening of the cerebellar fissures (blue arrows) and more atrophy of the superior vermis (yellow arrow), denoting progressive atrophy. The pons is also slightly smaller. Differences are more evident when comparing (**C**) with (**A**). FLAIR T2 also shows thinning of the superior cerebellar cortex with high signal intensity (red arrows) that might correspond to cortical gliosis. Green arrows indicate optic nerve atrophy. Despite clinical microcephaly, no other structural abnormalities were found.

**Figure 2 ijms-24-13699-f002:**
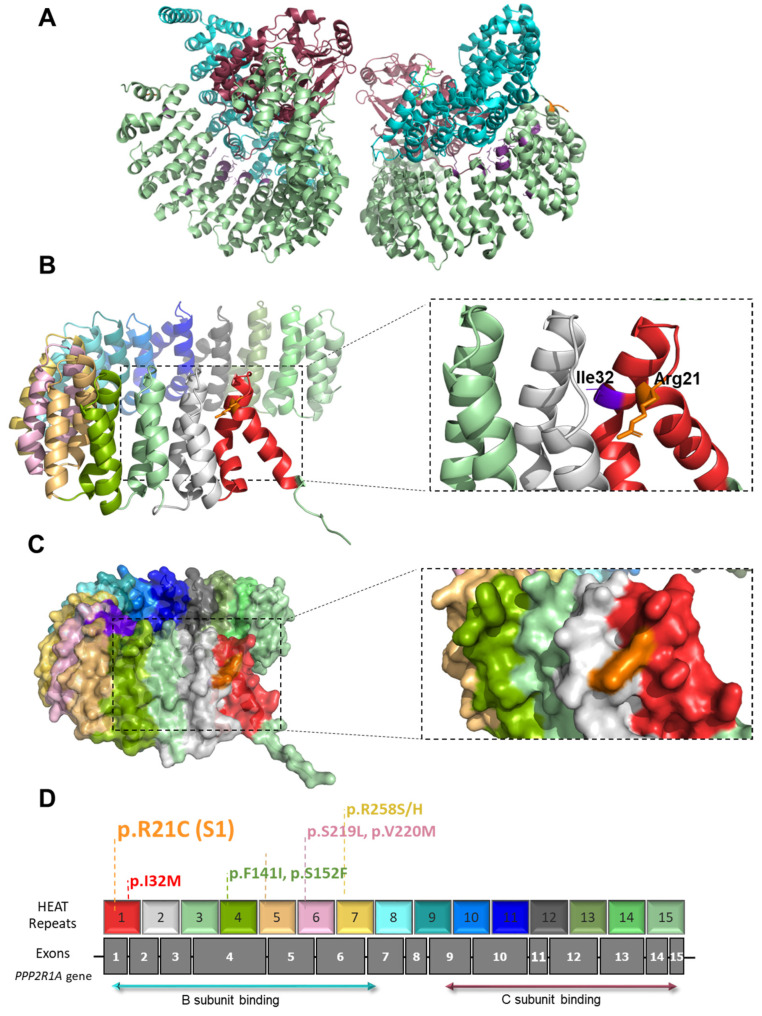
Three-dimensional visualization of the Protein Phosphatase 2A Holoenzyme model (2NPP [19]). (**A**) Overview of the full holoenzyme. The catalytic subunit is shown in red, the regulator in blue, and the scaffold subunit, coded by *PPP2R1A*, in green. The Arg21 residue is in orange and previously published pathogenic variants are depicted in purple (listed in Appendix A). (**B**) Details of the isolated PPP2R1A protein and the 15 HEAT domains (each HEAT domain is depicted in a different color and Arg21 is in orange), and AlphaFold model AF-P30153-F1. On the right, closeup cartoon representation of Arg21 (orange). Residue I32, in purple, previously reported, is shown on HEAT1-2 interface. (**C**) Overview of the protein surface with previously published pathogenic variants in purple and Arg21 in orange. (**D**) Gene representation with the previously reported variants and HEAT repeats, following the color code of (**B**).

**Figure 3 ijms-24-13699-f003:**
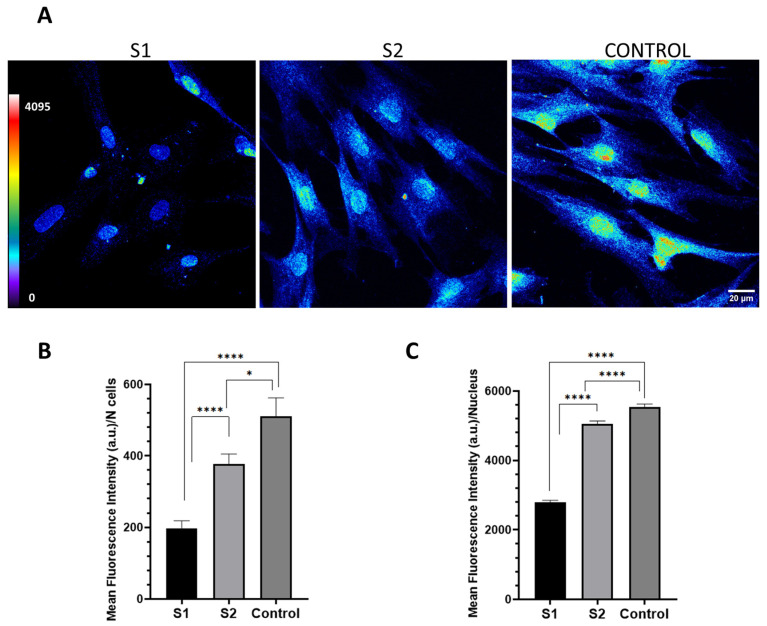
Confocal microscopy projections of fibroblasts illustrating the expression of *PPP2R1A*. (**A**) Representative pseudocolored images depict the abundance of PPP2R1A. The pseudocolor scale enables visual comparison, revealing lower fluorescence intensity (lower protein density) in the patients’ samples compared to the control. In particular, S1 exhibits minimal fluorescence signal in cells. Protein is observed both in the cytoplasm and the nucleus. The pseudocolor scale is shown in the bottom left corner of the figure. Warm colors, such as white and red, indicate high intensities, whereas cold colors, such as blue, represent low intensities. Scale bar: 20 µm. (**B**,**C**) Graphical representation of the intensity quantification studies using ImageJ/Fiji software ver. 1.52n (Wayne Rasband National Institutes of Health, Bethesda, MD, USA) The graphs depict global intensity normalized to the number of cells (**B**) and intensity localized in the nucleus (**C**). In both plots, S1 shows the lowest mean intensity value (arbitrary units, a.u.), while S2 and the control exhibit closer values. Specifically, in graph (**C**), the difference between S2 and the control is smaller compared to graph (B), suggesting similar expression of this protein in the nucleus. The data presented are the mean ± SEM. Significance levels are denoted as * *p* < 0.05 and **** *p* < 0.0001.

**Figure 4 ijms-24-13699-f004:**
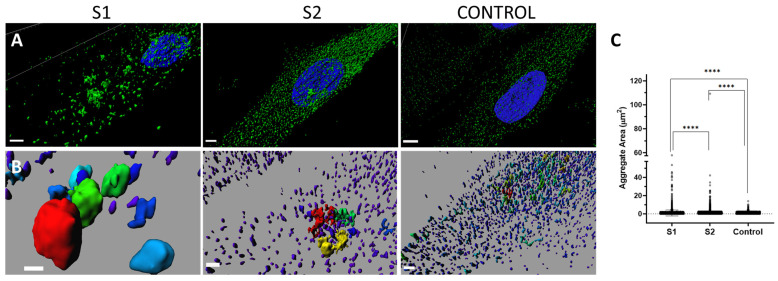
Representative super-resolution 3D images depicting *PPP2R1A* aggregates with variations in area and distribution. (**A**) Three-dimensional cell models were generated using Imaris software ver. 7.2.1 (Bitplane AG, Zürich, Switzerland), where protein aggregates are displayed in green and the nucleus in blue. S1 exhibits predominantly moderate and large aggregates, while S2 shows a higher concentration of small and moderate aggregates. The control displays primarily small aggregates. Scale bar: 3 µm. In (**B**), a pseudocolor spectral scale is used to indicate the area of different aggregates, with the larger aggregates appearing in red within each image. The larger aggregates, which vary in size across different samples (control and patient groups), are highlighted in a vibrant red hue, while the smaller aggregates are depicted in cooler shades, predominantly blue. Scale bar: 2 µm. (**C**) Graphical representation of the area occupied by aggregates in the 3D models of the subjects. S1 and S2 exhibit a higher presence of larger aggregates compared to the control group, with S1 showing larger aggregate areas than S2. The data, represent the mean ± SEM, and individual values are depicted as dots. Significance levels are denoted as **** *p* < 0.0001.

## Data Availability

The data supporting the reported results are available upon request.

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
