# Peer review of "Advanced Optical Microscopy: Unveiling Functional Insights Regarding a Novel *PPP2R1A* Variant and Its Unreported Phenotype"

_ijms, 2023, doi:10.3390/ijms241813699_

Round 1
Reviewer 1 Report
A novel Arg21Cys pathogenic variant in the protein PPP2R1A is described in this work. Magnetic resonance imaging revealed cerebellar asymmetric hypoplasia, pontocerebellar, and optic nerve atrophy, among others. Confocal microscopy of the affected cells revealed a decrease in the intensity of the PPP2R1A protein signal compared to those of cells with another PPP2R1A mutation, and healthy control. Computational super-resolution microscopy of these cells points to high molecular weight protein aggregates, some larger than 55 mm2. No such aggregates were detected in the cells with the wild-type protein.
The clinical significance of the findings in this study is evident. A major revision of the manuscript is suggested because there are several issues that need to be resolved before publication.
Major points:
- When describing the scan in Figure 1B, the authors claim that "…..signal reinforcement at the cerebellar cortex, denoting Purkinje cells gliosis. Supratentorial structures showed microcephaly…." It is difficult to appreciate the signal alterations - if any) present at the cerebellum on the image that can readily correspond to the "Purkinje cell gliosis". An arrow should be placed in the appropriate location if the authors are confident in their findings. It is difficult to evaluate microcephaly solely on MR images without additional clinical findings like head circumference measurements. Axial MR images are necessary for this purpose.
- The authors state in Figure 1C "…. cerebellar hemispheric atrophy progression with increased cortical signal….". They should put an arrow on the T2 WI to the abnormal signal area.
- Figure 2A and Figure 2B - There is no need to depict the entire protein; it may be sufficient to show the relevant HEAT1 domain with a close-up - possibly in the form of an inset) of the area near the mutation. The mutated residue should be clearly identified. In addition, figure 2B has been cropped in a noticeable manner.
- The authors should elaborate on the structural importance of an Arg21Cys mutation. Furthermore, work on how this substitution affects the PPP2R1A function - e.g. phosphate transfer experiments) would clarify the implications of this mutation.
- In Figure 3, nuclear staining and co-localization with PPP2R1A should be shown separately. Furthermore, it would be intriguing to observe a Western blot analysis of these samples using anti-PPP2R1A.
- The graphs are insufficiently explained throughout the manuscript. For example, the calculation of figure 4 A/B is not explained in detail.
- The quality of figure 5 has to be improved: - e.g. The authors used varying scale bars, there is no scale bar in the first image, and unequal representations of the aggregates in size)
- The authors correctly surmise that the accumulation of high molecular weight aggregates in the cells harbouring the Arg21Cys mutant results from the dysfunction of the protein degradation machinery. Functional experiments validating this claim would further strengthen this work as already described in the manuscript - e.g. page 9, line 332; increased oxidative stress?)
- The supplementary figure should be accompanied by the appropriate figure legend, outlining the PPP2R1A alignment and the conservation of a basic amino acid - Arg21 or Lys21) in this position among species.
- Due to the description in ClinVar, it would be noteworthy to contact Ambry Genetics, asking for details on the phenotype of the patient and about the origin of the variant. The authors should take this opportunity into account.
Minor points:
- The classification of ACMG should be revised because the PM1 criterion is not met - the mutation is not located in a hot spot, and there few benign missense variants near the described missense one. The authors themselves describe). The PS3 criterion is this study's object, and I wonder whether this should be included in the ACMG computation.
- The authors should specify the MRI pulse sequences used for the imaging.
- The authors should indicate product numbers for their used antibodies
- Age and sex of control fibroblasts should be indicated in the manuscript
- Figure 3 and 4 should be merged into one figure because this is just one experiment - same for figure 5 and 6).
- For analysis of Fig 4 + 6 the authors should use mean ± SD instead of mean ± SEM. The value of "****p" is not given throughout the manuscript.
- The labelling of axes in figure 4 A and B is inaccurate: - a.u.) is missing in 4A.
- Page 3, line 103: "S1" is not explained before
- Page 3, line 104: the bracket is not closed
- Page 3, line 128: no explanation for "WISC-IV"
- Page 4, line 188: "Supplementary material" - be more precise and indicate - e.g. Table S1, Figure S1)
- Page 6, Line 216 + 217: P1 and P2 were not explained - mixed up with S1 and S2 throughout the manuscript)
- Page 8, line 274: no explanation for HPC
- Page 9, line 286: no explanation for PCLO
- Page 11, line 412: should be "formaldehyde" instead of "paraformaldehyde"
- Page 11, line 427: Please indicate how many images were analyzed and quantified
Reviewer 2 Report
This study reports on an individual with an unreported de novo variant in PPP2R1A who presents with pontocerebellar hypoplasia (PCH), microcephaly, optic and peripheral nerve abnormalities, but without typical features like epilepsy and abnormal corpus callosum. The authors did comparative analyses of the patient's fibroblasts compared with different controls. Their functional analyses and microscopy revealed reduced PPP2R1A expression and abnormal protein aggregates, supporting the variant's pathogenicity. Overall, the manuscript presents valuable findings that expand the clinical spectrum of PPP2R1A-related neurodevelopmental disorders. I have the following comments to improve the manuscript.
1. Introduction: To date, there are three forms of Houge-Janssens syndrome (HJS) are caused by mutations in the protein phosphatase type 2 family of genes. The authors mentioned two of the three related genes, PPP2R1A and PPP2R5D, but did not introduce PPP2CA. It would be beneficial to include PPP2CA and the related neurodevelopmental disorder (MIM#618354) in the introduction to provide a comprehensive overview of the relevant genes and disorders.
2. MRI data: In the manuscript, it would be helpful to enhance the presentation of MRI abnormalities by using arrows to indicate the specific areas of interest, similar to how they indicated optic nerve atrophy with arrows.
3. Control cells information: Provide clear and essential information about the control cells, such as the age and gender of the donor, to establish a better context for the comparative analyses.
4. Figure 5, upper panel: The legend indicates that PPP2R1A protein aggregates are displayed in green. However, it is not clear if the green signal represents only the aberrant protein aggregates or if it also indicates the protein levels. The signal seems inconsistent with Figure 3 and Figure 4, which requires clarification. Ensure that this aspect is thoroughly explained to avoid any confusion.
Round 2
Reviewer 1 Report
The authors improved the manuscript as I suggested, although they did not do all the experiments, which I think could have improved the quality of the work even more.